# Increased Incidence and Clinical Features of Septic Arthritis in Patients Aged 80 Years and above: A Comparative Analysis with Younger Cohorts

**DOI:** 10.3390/pathogens13100891

**Published:** 2024-10-11

**Authors:** Hanna Alexandersson, Mats Dehlin, Tao Jin

**Affiliations:** 1Department of Rheumatology and Inflammation Research, Institute of Medicine, Sahlgrenska Academy, University of Gothenburg, 41346 Gothenburg, Sweden; mats.dehlin@vgregion.se (M.D.); tao.jin@rheuma.gu.se (T.J.); 2Department of Rheumatology, Sahlgrenska University Hospital, 41345 Gothenburg, Sweden

**Keywords:** septic arthritis, aging, incidence, synovial fluid, Sweden

## Abstract

Background: This study aimed to determine the incidence of septic arthritis across adult age groups in Västra Götaland Region (VGR) of Sweden, while also comparing disease characteristics among different age groups with hematogenous septic arthritis. Methods: Using ICD-10 codes for septic arthritis from 2016 to 2019, we identified 955 patients in VGR. We reviewed the medical records of 216 adult patients with hematogenous septic arthritis and compared data across age groups. Results: The overall incidence of septic arthritis in adults was 4 per 100,000 persons annually, rising to 14 per 100,000 in those ≥80 years. The median age of the 216 patients was 71. The comparison across age groups (18–64, 65–79, and ≥80) showed significantly longer hospital stays and higher mortality rate in the older groups. CRP levels were higher in the middle age group, SF-WBC counts were lower in the youngest age group, and synovial fluid crystals were more common in the oldest. No differences were found in joint involvement or the organisms isolated. Conclusion: The incidence of septic arthritis is 6.5 times higher in patients aged ≥ 80 compared to those under 65, highlighting the need to consider age-related differences in disease management.

## 1. Introduction

When a patient presents with one or, in rare cases, multiple acutely swollen and painful joints, septic arthritis must be ruled out promptly due to its high mortality and morbidity rates [1,2,3]. The diagnosis of septic arthritis is definitive if an organism is isolated from synovial fluid. However, in nearly 20% of cases, synovial fluid cultures remain negative even when septic arthritis is suspected [4], making the diagnosis challenging in clinical practice. Risk factors for septic arthritis include joint damage from previous trauma or arthritis caused by rheumatoid arthritis, osteoarthritis, or gout, as well as conditions like diabetes, skin infections, kidney failure, intravenous drug use, hemodialysis, and advanced age [5,6]. In all age and risk groups, *Staphylococcus aureus* is the most frequent causative organism, followed by other gram-positive bacteria, including *Streptococci* [7].

The incidence of septic arthritis has been generally reported to be low with a range of 4–10 cases per 100,000 persons annually [7,8]. These data are based on studies conducted in Australia by Morgan et al. 1996 [9], the Netherlands by Kaandorp et al. 1997 [10], and in Iceland by Geirsson et al. 2008 [11]. However, more recent studies from Thailand by Foocharoen et al. 2023 [12], from New Zealand by McBride et al. 2020 [13], and from the UK by Rutherford et al. 2016 [14] indicate a higher incidence, ranging between 6.7 and 23 per 100,000 persons annually. Notably, the studies from Iceland published in 2008 [11] and the study from the UK published in 2016 [14] observed a significant increase in the incidence during their study periods. The incidence of septic arthritis varies significantly with age, with higher rates observed in older populations. The incidence in the age group over 60 has been reported to be 57.7 per 100,000 persons annually by Foocharoen et al. [12]. Kennedy et al. [15] from New Zealand found an incidence of 58 per 100,000 persons annually in patients over 80. Rutherford et al. [14] reported an incidence of 31 per 100,000 persons annually for patients over 75. All these studies strongly suggest a correlation between advanced age and the increased incidence of septic arthritis.

In Sweden, there are no published estimates for the incidence of septic arthritis. This study aimed to fill the gap by determining the overall incidence of septic arthritis in Västra Götaland Region (VGR) of Sweden. Additionally, we sought to present the incidence rates across different age groups as well as to compare clinical findings and characteristics among these age groups. By analyzing these factors, this study provides valuable insights into the epidemiology of septic arthritis in this region, helping to identify age-related trends and improve clinical management for patients with septic arthritis.

## 2. Materials and Methods

### 2.1. Data Sources

#### 2.1.1. Data Collection to Estimate the Incidence of Septic Arthritis

Sweden is divided into six health care regions, with the Västra Götaland Region (VGR) representing approximately 20% of the Swedish population. VGR is considered to be representative of Sweden with regard to healthcare utilization, health status, and socioeconomics [16]. All ICD codes for various diseases in VGR are recorded in the Western Swedish Health Care Register (VEGA). In this study, data were collected from VEGA for all patients receiving ICD-10 codes for septic arthritis (M00.0–M00.9X/M01.0–M01.1X) between January 2016 and December 2019. The dataset included age at diagnosis and hospitalization days, resulting in a total of 955 patients.

In our previous study, the ICD-10 codes for septic arthritis were validated according to the modified Newman criteria [17], demonstrating a positive predictive value of 91% (unpublished data). Based on this, we assumed the same predictive value for the current study. Therefore, 91% of the 955 patients that were retrieved from VEGA were included in estimating the incidence of septic arthritis. Population data for VGR from January 2016 to December 2019, including the proportions of the population in different age groups (18–30 years, 31–64 years, 65–79 years, and over 80 years), were obtained from Statistics Sweden [18].

#### 2.1.2. Data Collection to Describe Characteristics of the Study Population

Of the 955 patients with an ICD-10 code for septic arthritis, we further investigated patients treated at two different hospitals in the region. We included one representative hospital from each category: NÄL-Uddevalla Hospital Group (NU) in Trollhättan and Uddevalla (county hospital), as well as Sahlgrenska University Hospital (SU) in Gothenburg to ensure consistency between university and county hospitals. Four hundred and seventy-five patients were treated for septic arthritis in these two hospitals during the study period from January 2016 to December 2019.

By reviewing the medical records, we excluded patients whose infection was due to direct inoculation (e.g., postoperative within 3 months or post-intraarticular injection) and included only those with a hematogenous route of infection. The reason for this exclusion was that the hematogenous spread septic arthritis and the direct inoculation groups differed in terms of the type of organism involved and the clinical presentation [6]. In addition, the age distribution between these two groups differed significantly (Appendix A), with hematogenous septic arthritis contributing most to the increased incidence of the disease in older individuals.

We collected data on organisms isolated from blood, synovial fluid, or synovial tissue, identified through culture or polymerase chain reaction (PCR) analysis. This resulted in a study population of 216 patients (Figure 1) with hematogenous spread septic arthritis, in which an organism had been identified. Additionally, we recorded any observed antimicrobial resistance patterns to provide a comprehensive understanding of the microbial landscape and resistance issues in these cases.

Data collected from the medical records included patient age, sex, hospitalization duration, number of affected joints, site of the infection, and mortality within 30 days of diagnosis. The body temperature was measured either at the emergency ward or within the first 24 h after admission. Blood sample results at admission were also documented, including hemoglobin (Hb) in g/L, platelets (×10^9^/L), C-Reactive Protein (CRP) in mg/L, creatinine in µmol/L, glucose level (mmol/L), and white blood cell count (WBC × 10^9^/L). Furthermore, results from the synovial fluid analysis were collected, including glucose level (mmol/L), synovial WBC (SF-WBC × 10^9^/L), polymorphonuclear neutrophils (PMNs × 10^9^/L), and the presence of crystals (both urate and pyrophosphate). The glucose ratio was calculated by dividing the glucose level in peripheral blood by the synovial fluid glucose level, and the SF-PMN/WBC ratio was determined by dividing the SF-PMNs by the SF-WBC.

### 2.2. Statistical Analyses

Descriptive statistics were used to summarize the demographics and characteristics of the total patient population. To calculate the incidence, the population was divided into four age groups: 18–30 years, 31–64 years, 65–79 years, and over 80 years. However, to compare demographics and characteristics, the population was divided into three age groups: 18–64, 65–79, and over 80 years. These age groups were chosen to ensure a balanced number of patients in each group, as only eight patients were aged 18–30 years.

For comparisons between these age groups, One-way ANOVA, Chi-square test, or Fisher’s exact test was used when appropriate. When significance was identified, independent samples *t*-tests were conducted to compare each age group with the others. Univariable and multivariable logistic regression models were used to assess the associations of the three age groups with CRP, SF-WBC, and hospitalization days after adjustments for sex, obesity (BMI ≥ 30), type of organism, joint location, and comorbidities: diabetes type 1 and 2, osteoarthritis, dialysis, juvenile idiopathic arthritis, alcohol liver disease, malign cancer ongoing or treated, kidney failure, psoriatic arthritis, and rheumatoid arthritis. The level of significance was set at *p* < 0.05. Statistical analyses were performed using IBM SPSS Statistics for Windows, Version 28 (released 2021; IBM Corp., New York, NY, USA) and SAS version 9.4 software (SAS Institute Inc. Cary, NC, USA). Sensitivity and specificity were not calculated because this would have required a second population to determine the proportions of false and true negatives. Ethical approval was received from the Ethical Review Board of Gothenburg, Sweden.

## 3. Results

### 3.1. Incidence of Septic Arthritis in Västra Götaland Region (2016–2019)

The VEGA database identified 955 patients classified with at least one of the ICD-10 codes for septic arthritis between January 2016 and December 2019. Given that the ICD-10 codes for septic arthritis have a positive predictive value (PPV) of 91% in relation to the modified Newman criteria in Swedish inpatient care (unpublished data), we assumed that 91% of the 955 patients had the correct diagnosis, resulting in 869 patients. According to Statistics Sweden [18], the population of VGR during the four-year study period was 5.381859 million. This resulted in an overall incidence of septic arthritis in VGR between January 2016 and December 2019 of 4.0 per 100,000 persons annually.

In the age group 18–30 years with a population of 1.174762 million, 51 patients were diagnosed, resulting in an incidence of 1.1 per 100,000 persons annually. In the age group 31–64 years, with a population of 2.890925 million, 292 patients were diagnosed, leading to an incidence of 2.5 per 100,000 persons annually. In the age group 65–79 years with a population of 0.972398 million, 333 patients were diagnosed, resulting in the calculated incidence of 8.5 per 100,000 persons annually. In the age group over 80 years with a population of 0.343774 million, 193 patients were diagnosed, resulting in a calculated incidence of 14 per 100,000 persons annually. The incidence in the age group over 80 was nearly 13 times higher than in the age group 18–30 years (Figure 2).

### 3.2. Patient Characteristics

The study population consisted of 216 patients with hematogenic septic arthritis identified between January 2016 and December 2019. The demographics are detailed in Table 1. Of the total group, 58% were men, with a median age of 71 years (range 19–96). The median hospitalization duration was 17 days (range 1–80 days), and the mortality rate within 30 days after diagnosis was 5% (11 patients).

Fever (≥37.5 Celsius) during the first 24 h after arrival at the emergency ward was observed in 114 (54%) of the patients. The mean CRP was 222 mg/L ± 129 mg/L, with CRP ≤ 10 mg/L in 4 cases (Table 1). Additional blood sample results showed a mean WBC of 15 × 10^9^/L ± 18 × 10^9^/L, mean platelets 253 × 10^9^/L ± 131 × 10^9^/L, and mean creatinine 127 µmol/L ± 111 µmol/L.

The review of the medical records revealed that synovial fluid analysis was only performed in 71–79 (33–37%) of the patients due to issues such as insufficient fluid volume, aspiration difficulties, or coagulation in the test (Table 1). To determine whether there was a significant difference between patients who underwent synovial fluid analysis and those who did not, we compared the two groups and found no significant differences (Appendix A).

The mean SF-WBC was 64 × 10^9^/L ± 51 × 10^9^/L, the mean glucose ratio was 0.36 ± 0.31 mmol/L, and the mean SF-PMN/WBC ratio was 0.88 ± 0.13. Crystals (urate or pyrophosphate) in synovial fluid were analyzed in 79 of the patients (37%), of whom 18 patients (23%) had crystals in the synovial fluid, despite a definitive diagnosis of septic arthritis confirmed by the isolation of an organism from the blood or the affected joint (synovial fluid or tissue).

### 3.3. Comparison between Different Age Groups

When comparing the three different age groups (18–64 years, 65–79 years, and over 80 years), we found that the hospitalization days were significantly longer for patients over 65 years compared to younger patients (*p* < 0.05), and the mortality rate was significantly higher in the age group over 80 years compared to the youngest age group (*p* < 0.001). There was no significant difference in the frequency of fever or the occurrence of mono/oligoarthritis between the age groups (Table 1).

In blood sample comparisons, we found significantly higher CRP in the age group 65–79 years compared to the youngest (*p* < 0.01), a difference that was still significant after adjustments were made for sex, type of organism, joint location, and comorbidities in a multivariable logistic regression analysis (Table 2). Furthermore, CRP was higher in the age group 65–79 years compared to the oldest group (*p* < 0.05), but this difference was no longer significant after adjustments were made for sex, type of organism, joint location, and comorbidities in a multivariable regression analysis (Table 2). Hemoglobin levels were significantly lower in the age group of 80 and above compared to the youngest group (*p* < 0.01). No significant differences were found between the age groups in WBC, platelets, and creatinine levels (Table 1).

As for the analysis of synovial fluid, we found a significant difference in the SF-WBC with the lowest in the age group 18–64 years compared to the age group 65–79 years (*p* < 0.05, Table 1), and this significance was still present after adjustments were made for sex, type of organism, joint location, and comorbidities in a multivariable regression analysis (Table 2). Furthermore, crystals in the synovial fluid were significantly more common in the age group 80 years and above compared to the two other groups (*p* < 0.01). Notably, 52% of the patients aged 80 years and above had crystals in SF despite a confirmed septic arthritis diagnosis with an isolated organism. No significant difference was found when comparing the SF-PMN/WBC ratio or glucose ratio between groups (Table 1).

### 3.4. Joint Involvement in Different Age Groups

To determine whether joint involvement differed between age groups, we analyzed the distribution of the affected joints in our study population. Overall, the majority of patients had monoarthritis (94%). Among these cases, the joint distribution was as follows: knee (41.2%), glenohumeral joint (11.6%), hip (9.0%), and wrist (6.5%). When comparing the different age groups, there were no significant differences in the distribution of the affected joints or the frequency of mono/oligoarthritis in septic arthritis (Table 3).

### 3.5. Isolated Organisms

Among the 216 patients, mixed infections were present in 15 patients (7%), resulting in a total of 231 isolated organisms (Table 4). *Staphylococcus aureus* was the most common pathogen, identified in 116 cases (50%), followed by β-hemolytic *Streptococci* in 48 (21%) cases. There was no significant difference between the distribution of organisms causing septic arthritis across the three age groups (Table 3). Furthermore, only one patient had a resistant species, extended spectrum beta-lactamase (ESBL), found in the age group of 80 years and above. Notably, no case of methicillin-resistant *Staphylococcus aureus* (MRSA) was detected in the study population.

## 4. Discussion

This study is the first to report the incidence rates of septic arthritis across different age groups in Sweden. The overall incidence of septic arthritis among adults in VGR between 2016 and 2019 was 4 per 100,000 persons annually. Previous studies have reported the incidence ranging from 4 to 23 per 100,000 persons annually [1,3,7,8,12,13,14,19]. These variations are likely due to differences in the countries studied, whether both children and adults or only adults were included, and the diagnostic criteria used.

To our knowledge, there are only a few studies presenting differences in the incidence of septic arthritis between age groups. Rutherford et al. [14] reported an incidence of septic arthritis in the UK over the period from 1998 to 2013 of 5.5 to 7.8 per 100,000 persons annually. They found that the incidence was significantly higher in individuals over 75 years, reaching 31 per 100,000 persons annually in 2013 [14]. The present study showed an incidence of 14 per 100,000 persons annually in those aged 80 and above. One notable difference between the two studies is our exclusion of 9% of patients based on a prior validation study of the ICD-10 codes in relation to septic arthritis using the modified Newman criteria (unpublished data). Another difference is that we excluded children, whilst Rutherford et al. included children. Additionally, we had a cut-off of 80 years instead of 75 years. However, these differences alone may not fully explain the significant difference in incidence rates between the two studies.

Kennedy et al. [15] conducted a similar study in New Zealand, reporting an overall incidence rate of 12 per 100,000 persons annually between 2009 and 2013. In the 80–84 age group, the incidence was 40 per 100,000 persons annually [15]. Despite the differences in incidence rates across countries, all studies indicate a higher incidence of septic arthritis among elderly people compared to a younger population [14,15]. Moreover, there appears to be a rising trend in overall incidence across all adult age groups over time [14]. This increase is likely due to factors such as aging populations, a rise in medical comorbidities, more frequent orthopedic procedures, increased use of immunosuppressive therapy, and prevalence of underlying joint diseases, such as rheumatoid arthritis, crystal arthropathies, and osteoarthritis [3,7,15].

Given the distinctly higher incidence of septic arthritis in elderly people [14,15] and the aging population in Europe, it is of great importance to analyze potential age-related differences in septic arthritis. In our study, with a study population of hematogenous spread septic arthritis, we found that the hospitalization days were significantly higher in patients over 65 years compared to younger patients. Furthermore, the 30-day mortality rate was significantly higher in patients aged 80 years and above compared to that in patients under 65 years. These findings are expected, considering the higher comorbidity burden and reduced infection resistance in older individuals. Indeed, recent experimental data also suggest that aging significantly increases the mortality rate in mice with *Staphylococcus aureus* bacteremia, independently of Toll-like Receptor 2 (TLR-2) [20]. Furthermore, lower hemoglobin levels in older population, as reported in previous studies [21], were also confirmed in the present study.

Blood sample results revealed a significant difference in mean CRP with significantly higher CRP in the middle age group (65–79 years) compared to the youngest age group. A large study from China with 4418 healthy participants showed a significant correlation between increasing CRP and advanced age [22]. However, that study only reports the CRP levels in healthy individuals and does not address how the CRP levels respond to infection across different age groups. In our study, there were no significant differences in the organisms causing septic arthritis between the age groups, indicating that the difference in CRP cannot be attributed to variations in infecting organisms. Furthermore, the difference in CRP levels could not be explained by differences in the distribution of affected joints among age groups (Table 2). We did observe that the presence of osteosynthesis or prosthesis material in the affected joint was more common in the middle-aged group compared to the youngest age group (Table 1), although it is unclear if this is related to the increased CRP levels. Nevertheless, many unknown confounding factors could contribute to the difference in CRP between the age groups, and further studies are needed to explore these factors.

Synovial fluid analysis is crucial when septic arthritis is suspected. In the present study, we found significantly lower mean SF-WBC of 44 × 10^9^/L ± 34 × 10^9^/L in the younger age group (18–64 years) compared to 77 × 10^9^/L ± 57 × 10^9^/L in the 65–79 years group (Table 1). This finding is based on 71 patients (33% of the study population) who underwent SF-WBC analysis. Guidelines typically suggest that SF-WBC exceed 50 × 10^9^/L when arthritis is caused by infection [23]. However, in our study, the youngest age group had a mean SF-WBC of 44 × 10^9^/L. This suggests that the level of SF-WBC in septic arthritis may vary by age group, an area that warrants further investigation.

Crystals in the synovial fluid are an important diagnostic tool for differentiating infectious arthritis from crystal arthropathy [6]. Our study, consistent with others, has highlighted that the presence of crystals in aspirates does not rule out the possibility of septic arthritis [15,24]. In the age group of 80 years and above, concomitant crystals in synovial fluid were present in 52% of the patients, significantly more common than in younger age groups. This finding emphasizes the importance of not dismissing the diagnosis of septic arthritis solely based on the presence of crystals, particularly in older patients.

The knee is the primary site affected by bacterial septic arthritis [2,6,8], a finding confirmed in the present study. Intriguingly, similar patterns have been observed in mouse models of hematogenous septic arthritis, where the knee joint is the most commonly affected site [25]. Understanding the underlying mechanism behind the preference of *Staphylococcus aureus* for targeting the knees warrants further investigations.

In this study, we found no significant differences in the distribution of affected joints in septic arthritis across age groups, nor in the frequency of multiple joint involvement or monoarthritis (Table 2). *Staphylococcus aureus* was the most common organism causing septic arthritis, accounting for 50% of infections (Table 4), consistent with previous studies [3,6,7,8]. However, some studies suggest that gram-negative organisms, such as *Escherichia coli,* could be more frequently found in elderly patients [26,27,28]. Our study found no significant differences in the distribution of organisms in the age groups, including *Escherichia coli* (Table 4). Of importance, our study did not identify any cases of methicillin-resistant *Staphylococcus aureus* (MRSA) associated with septic arthritis over the study period. This suggests a success in managing and controlling MRSA infections during the study period in Sweden.

Our study has several limitations because of its retrospective design. Furthermore, one potential issue is the risk of errors in estimating the incidence of septic arthritis, which could arise from false positive diagnoses, either due to administrative coding errors or diagnostic inaccuracies in patients not fulfilling the Newman criteria. To reduce the likelihood of false-positive diagnoses, we excluded 9% of the patients based on our previous study’s findings (unpublished data). Another limitation is the incomplete availability of data for certain parameters, such as synovial fluid analysis, across the study population. SF-WBC, for example, was only available in 74 (34%) of the patients in the study population. Our analysis comparing patients with available SF-WBC data to those without showed no significant differences (Appendix A), suggesting that the missing data may not bias the results. Furthermore, the exclusion of the direct inoculation group limits the generalizability of the findings to the broader septic arthritis population, making them only applicable to hematogenous spread septic arthritis. In addition, the youngest age group has a large age span from 18 to 64, which could potentially obscure trends or differences between younger adults and middle-aged adults. Despite this, we chose to cluster ages 18–30 and 31–64 to ensure a balanced and sufficient number of patients in each group, as the number of patients younger than 31 years was very small (Appendix A). However, a key strength of our study is the inclusion of only patients with isolated organisms, ensuring that our conclusions are drawn from cases with a definitive diagnosis. This approach significantly enhances the reliability of our findings despite the limitations mentioned above.

## 5. Conclusions

This study demonstrates that, although septic arthritis remains relatively rare, the overall incidence was 4 per 100,000 persons annually in VGR, Sweden between the years 2016 and 2019. The incidence in the age group of 80 years and above was approximately 6.5 times higher (14 per 100,000 persons annually) compared to patients under 65 years (2.1 per 100,000 persons annually. With an aging population, we can expect an increased rate of septic arthritis in the future. Septic arthritis remains a diagnostic challenge, as evidenced by the inconclusive tests and examinations highlighted in the present study. We also identified significant differences among age groups in the study population of hematogenous septic arthritis, including CRP levels, 30-day mortality, synovial fluid WBC counts, and the presence of synovial fluid crystals, which should be considered in daily practice when managing patients with potential septic arthritis. Further studies into these age-related differences are essential to improve the diagnosis and improve outcomes for patients with septic arthritis.

## Figures and Tables

**Figure 1 pathogens-13-00891-f001:**
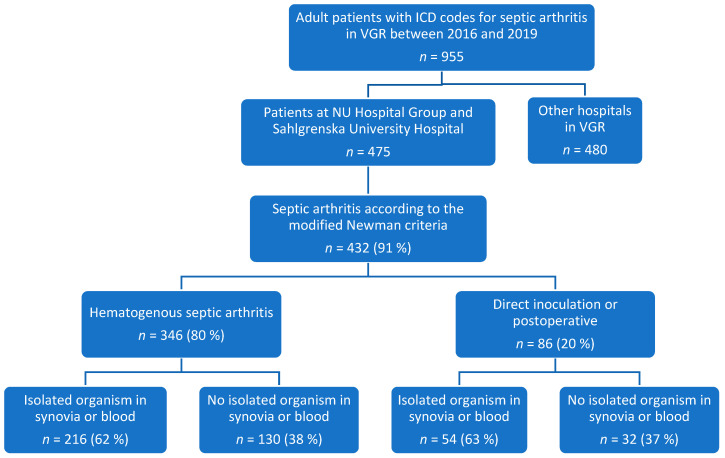
Diagram outlining the procedures for selecting the study population used to describe clinical data and compare age groups in patients with hematogenous septic arthritis. The diagram details the exclusion of patients with infections due to direct inoculation (e.g., direct inoculation or post-operative within 3 months) and the inclusion of those with hematogenous spread, along with the identification of organisms.

**Figure 2 pathogens-13-00891-f002:**
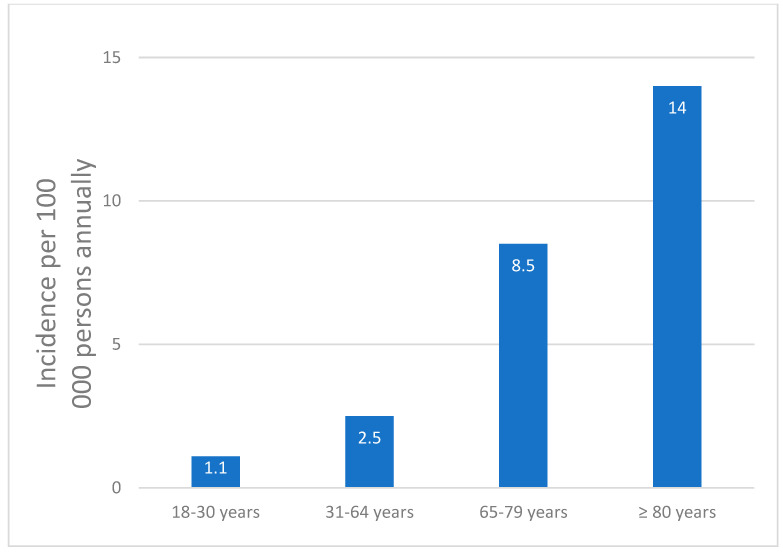
Incidence of septic arthritis per 100,000 persons annually in the Västra Götaland Region, Sweden, from 2016 to 2019, categorized by different age groups.

**Table 1 pathogens-13-00891-t001:** Demographics and disease characteristics of patients with hematogenous septic arthritis verified with isolated organisms at NU Hospital Group and Sahlgrenska University Hospital between 2016 and 2019.

Parameter	Total *n* = 216	1: Age 18–64 Years (*n* = 78)	2: Age 65–79 Years (*n* = 85)	3: Age ≥ 80 Years (*n* = 53)
Male (%)	125 (58)	44 (56)	53 (62)	28 (53)
Hospitalization days median (range)	17 (1–80)	13 (1–80) ^α^*	20 (2–61)	21 (5–57) ^β^*
Mortality within 30 days (%)	11 (5)	0 (0)	4 (5)	7 (13) ^β^***
Osteosynthesis or prosthesis in affected joint	29 (13)	5 (6) ^α^**	18 (21)	6 (11)
Fever ≥ 37.5 Celsius (%)	*n* = 210 114 (54)	*n* = 75 45 (60)	*n* = 82 43 (52)	*n* = 53 26 (49)
CRP mg/L mean (SD)	*n* = 211 222 (129)	*n* = 76 195 (137) ^α^**	*n* = 82 256 (123) ^γ^*	*n* = 53 210 (117)
Haemoglobin g/L mean (SD) *n* = 201	*n* = 201 123 (19)	*n* = 68 127 (19)	*n* = 80 122 (22)	*n* = 53 118 (16) ^β^**
WBC × 10^9^/L mean (SD)	*n* = 207 15 (18)	*n* = 74 13 (7)	*n* = 80 15 (22)	*n* = 53 16 (22)
Platelets × 10^9^/L mean (SD)	*n* = 187 253 (131)	*n* = 66 279 (139)	*n* = 69 243 (131)	*n* = 52 234 (117)
Creatinine µmol/L mean (SD)	*n* = 201 127 (111)	*n* = 69 109 (105)	*n* = 80 145 (131)	*n* = 52 125 (82)
SF-WBC × 10^9^/L mean (SD)	*n* = 74 64 (51)	*n* = 25 44 (34) ^α^*	*n* = 27 77 (57)	*n* = 22 72 (52) ^β^*
Glucose ratio mmol/L (SD)	*n* = 71 0.36 (0.31)	*n* = 24 0.38 (0.31)	*n* = 27 0.32 (0.32)	*n* = 20 0.40 (0.28)
SF-PMN/WBC ratio (SD)	*n* = 72 0.88 (0.13)	*n* = 24 0.87 (0.11)	*n* = 27 0.87 (0.16)	*n* = 21 0.90 (0.11)
SF-Crystals (%) Total Pyrophosphate Urate ^a^	*n* = 79 19 (24) 11 (14) 8 (10)	*n* = 27 3 (11) 1 (4) 2 (7)	*n* = 29 4 (14) ^γ^** 3 (10) 1 (4)	*n* = 23 12 (52) ^β^** 7 (30) 5 (22)

^a^ One patient had both urate and pyrophosphate crystals. For the statistics, one-way ANOVA was used. The Chi^2^ test (Pearson Chi-Square) was used for statistics on hospitalization days. Fisher’s exact test was used for statistics on SF crystals and mortality within 30 days. * *p* ≤ 0.05, ** *p* ≤ 0.01, *** *p* ≤ 0.001. ^α^, ^β^, and ^γ^ represent statistical differences evaluated with the independent samples *t*-test with significant *p* < 0.05. ^α^: comparison of groups 1 and 2; ^β^: comparison of groups 1 and 3; ^γ^: comparison of groups 2 and 3.

**Table 2 pathogens-13-00891-t002:** Uni- and multivariate logistic regression analyses on levels of CRP, SF-WBC, and hospital duration stratified by age groups.

Age Groups	Variable Value, Mean (SD)	OR ^a^, Univariate	*p*-Value	OR, Multivariable ^b^	*p*-Value
CRP, mg/L					
18–64, ref	195 (165)	1		1	
65–79	256 (253)	2.5 (1.4–4.4)	0.001	4.4 (2.3–8.4)	<0.0001
≥80	210 (212)	1.4 (0.8–2.6)	0.3	2.3 (11–5.0)	0.03
SF-WBC, ×10^9^/L					
18–64, ref	44 (38)	1		1	
65–79	77 (65)	3.1 (1.2–8.2)	0.02	8.8 (2.4–32.2)	0.001
≥80	72 (56)	2.7 (0.97–7.4)	0.057	1.4 (0.3–6.6)	0.6
Hospital duration, days					
18–64, ref	16 (13)	1		1	
65–79	22 (20)	3.1 (1.8–5.3)	<0.0001	3.5 (1.8–6.6)	0.0001
≥80	21 (21)	2.9 (1.6–5.4)	0.0008	2.5 (1.2–5.4)	0.02

^b^ OR: odds ratio. ^a^ Multivariable regression adjusted for sex, type of organism, joint location, diabetes type 1 and type 2, osteoarthritis, dialysis, obesity (BMI ≥ 30), juvenile idiopathic arthritis, alcoholic liver disease, malign cancer ongoing or recent, kidney failure, psoriatic arthritis, and rheumatoid arthritis.

**Table 3 pathogens-13-00891-t003:** Location of septic arthritis in relation to age groups.

Infected Joint	Total *n* = 216	Age 18–64 Years (*n* = 78)	Age 65–79 Years (*n* = 85)	Age ≥ 80 Years (*n* = 53)	*p*-Value
Knee	89 (41.2)	28 (35.9)	39 (45.9)	22 (41.5)	NS
Glenohumeral	25 (11.6)	7 (9.0)	9 (10.6)	9 (17.0)	NS
Hip	20 (9.3)	5 (6.4)	10 (11.8)	5 (9.4)	NS
Wrist	14 (6.5)	4 (5.1)	7 (8.2)	3 (5.7)	NS
Hand	13 (6.0)	8 (10.3)	4 (4.7)	1 (1.9)	NS
Back	13 (6.0)	4 (5.1)	5 (5.9)	4 (7.5)	NS
Ankle	11 (5.1)	7 (9.0)	2 (2.4)	2 (3.8)	NS
Other ^a^	7 (3.2)	3 (3.8)	2 (2.4)	2 (3.8)	NS
Foot	6 (2.8)	3 (3.8)	1 (1.2)	2 (3.8)	NS
Elbow	6 (2.8)	2 (2.6)	2 (2.4)	2 (3.8)	NS
Multiple joint involvement	12 (5.6)	7 (9.0)	4 (4.5)	1 (1.8)	NS

^a^ Sternoclavicular *n* = 6 and symphysitis *n* = 1. The Chi^2^ test was used for statistics within the knee, and Fisher’s exact test was used for all the other joints compared to the age group. NS: not significant.

**Table 4 pathogens-13-00891-t004:** Organisms isolated, comparison between age groups.

Organism	Total *n* = 231	Age 18–64 Years (*n* = 82)	Age 65–79 Years (*n* = 93)	Age ≥ 80 Years (*n* = 56)	*p*-Value
*Staphylococcus aureus*	116 (50)	46 (56)	39 (42)	31 (55)	NS
β-hemolytic *Streptococci*	48 (21)	20 (24)	21 (23)	7 (13)	NS
Gram-negative rods	23 (10)	6 (7)	13 (14)	4 (7)	NS
*Streptococcus pneumoniae*	14 (6)	2 (2)	7 (8)	5 (9)	NS
Others ^a^	10 (4)	4 (5)	5 (5)	1 (2)	NS
*Coagulase-negative staphylococci*	10 (4)	1 (1)	5 (5)	4 (7)	NS
Other streptococcal species	10 (4)	3 (4)	3 (3)	4 (7)	NS

Fisher’s exact test was used for all the statistics except for *Staphylococcus aureus* where the chi^2^ test was used. ^a^ *Cutibacterium acnes*, *Propionibacterium, Enterococcus faecalis*. Mixed infections: *n* = 15, with 2 organisms: *n* = 11, with 3 organisms: *n* = 2. NS: not significant.

## Data Availability

The original contributions presented in the study are included in the article/Appendix A, further inquiries can be directed to the corresponding author.

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
