# Peer review of "Increased Incidence and Clinical Features of Septic Arthritis in Patients Aged 80 Years and above: A Comparative Analysis with Younger Cohorts"

_pathogens, 2024, doi:10.3390/pathogens13100891_

Round 1
Reviewer 1 Report
Comments and Suggestions for Authors
Your article is well written with helpful informations for doctors and patients involved in this area. I encourage you to continue the investigations in this field in order to obtain better outcome for this pathologies .
Reviewer 2 Report
Comments and Suggestions for Authors
The article presented for review addresses an important aspect, which is the increased incidence and clinical symptoms of septic arthritis in patients over 80 years of age compared to younger cohorts.
Abstract: The aim of the study is to determine the incidence of septic arthritis in adults in the Västra Götaland region of Sweden (VGR), and to compare the disease characteristics in different age groups with septic arthritis of hematogenous origin. Then, the study method, results and conclusions are presented.
The scope of keywords is correct, including reference to all issues included in the retrospective comparative analysis.
In the introduction, the authors explain that septic arthritis remains a rare disease entity. However, in patients with multiple swollen and painful joints, septic arthritis should be quickly excluded due to its high mortality. They emphasize that the diagnosis of septic arthritis is not easy in clinical practice, and synovial fluid cultures remain negative even when septic arthritis is suspected. They then present epidemiological data on septic arthritis in several countries. They note that the incidence of septic arthritis varies considerably by age, with higher rates observed in older populations.
Methods: Using ICD-10 codes for septic arthritis from 2016 to 2019, the authors identified 955 patients in the VGR (approximately 20% of the Swedish population). ICD-10 codes were validated according to modified Newman criteria. The authors reviewed the medical records of 216 adult patients with septic arthritis of hematogenous origin and compared the data across age groups. This section details the principles of patient data collection (Figure 1) and statistical analysis methods. The study results, including the characteristics of the patient group, are presented in Figure 2 and Tables (1,2,3).
Results: The authors showed that the comparison across age groups (18-64, 65-79 and over 80) showed significantly longer hospital stays and higher mortality rates in the older groups. CRP levels were higher in the middle-aged group, and synovial fluid crystals were more common in the oldest.
Discussion: The authors clearly compare and interpret their study results in correlation with studies by other authors. Despite differences in the incidence of septic arthritis in different countries, all studies indicate a higher incidence of the disease in the elderly compared with the younger population. Staphylococcus aureus was the most common organism causing septic arthritis, accounting for 50% of infections (Table 4? rather Table 3), which is consistent with the results of the authors' previous studies.
Conclusions: The authors conclude that the incidence of suppurative arthritis is six times higher in patients over 80 years of age compared with patients under 65 years of age, emphasizing the need to take into account age-related differences in the management of the disease. Furthermore, septic arthritis remains a diagnostic challenge, as evidenced by the equivocal tests and investigations that should be considered in everyday practice when treating patients with potential septic arthritis.
The article uses 28 current scientific literature sources - (all cited).
This original article on septic arthritis conducted by retrospective analysis of medical records of patients with rheumatological conditions is an important first study on the incidence of septic arthritis in the Swedish Västra Götaland region (VGR). It is also significant that the authors' study did not identify any cases of methicillin-resistant Staphylococcus aureus (MRSA) associated with septic arthritis during the study period. This suggests good control of MRSA infections during the study period in Sweden.
In my opinion, the research process presented in the article is reliable and correct. I enjoyed reading this article and recommend it for publication.
Reviewer 3 Report
Comments and Suggestions for Authors
1. L28: The diagnosis "of" septic arthritis
2. L85: The authors excluded patients with direct inoculation but only analyzed those with hematogenous septic arthritis. It would be beneficial to elaborate more on the exclusion criteria and potential biases introduced by focusing only on this type of infection. This could limit the generalizability of the findings, especially comparing with other studies. Advanced aged population also tend to develop hematogenous infection, which could bias the result. Please elaborate it.
3. Table 1: The synovial fluid analysis was only performed in less than 50% of patients due to various issues. This limits the study’s ability to generalize the findings related to SF-WBC and SF-PMN counts. A clearer discussion on how these missing data points may bias the results is necessary. Additionally, the description of reasons for the incomplete availability of synovial fluid analysis lacks detail. A sensitivity analysis comparing patients with and without synovial fluid analysis could provide insights into whether these missing data could have altered the findings.
4. The study compares three age groups (18-64, 65-79, and over 80). However, the rationale for grouping ages 18-64 as one group is not adequately explained, especially considering the wide variability in immune function and joint health between these ages. This wide age group may obscure specific trends or differences in younger adult populations compared to middle-aged adults.
5. The manuscript reports several associations between age and clinical outcomes (e.g., CRP, hospitalization length), but all comparisons appear to be univariate (ANOVA, t-tests, etc.). Since factors such as comorbidities, organism type, and joint involvement could confound these associations, a multivariable regression model would improve the robustness of the findings.
6. The authors report significant differences in CRP levels and SF-WBC counts between age groups. However, they do not adjust for potential confounders such as underlying health conditions, which are likely more prevalent in the older population and could influence these laboratory parameters.
Comments on the Quality of English Language
No comment.
Round 2
Reviewer 3 Report
Comments and Suggestions for Authors
The revised version addressed all the issues I mentioned. I have no further comment.